# Antimicrobial and Phylogenomic Characterization of *Bacillus cereus* Group Strains Isolated from Different Food Sources in Italy

**DOI:** 10.3390/antibiotics13090898

**Published:** 2024-09-20

**Authors:** Donatella Farina, Angelica Bianco, Viviana Manzulli, Stefano Castellana, Antonio Parisi, Marta Caruso, Rosa Fraccalvieri, Luigina Serrecchia, Valeria Rondinone, Lorenzo Pace, Antonio Fasanella, Valerio Vetritto, Laura Maria Difato, Dora Cipolletta, Michela Iatarola, Domenico Galante

**Affiliations:** Istituto Zooprofilattico Sperimentale della Puglia e della Basilicata, Via Manfredonia 20, 71121 Foggia, Italy; angelica.bianco@izspb.it (A.B.); viviana.manzulli@izspb.it (V.M.); stefano.castellana@izspb.it (S.C.); antonio.parisi@izspb.it (A.P.); marta.caruso@izspb.it (M.C.); rosa.fraccalvieri@izspb.it (R.F.); luigina.serrecchia@izspb.it (L.S.); valeria.rondinone@izspb.it (V.R.); lorenzo.pace@izspb.it (L.P.); antonio.fasanella@izspb.it (A.F.); valerio.vetritto@izspb.it (V.V.); maria.difato@izspb.it (L.M.D.); dora.cipolletta@izspb.it (D.C.); michela.iatarola@izspb.it (M.I.); domenico.galante@izspb.it (D.G.)

**Keywords:** antibiotic resistance, *Bacillus cereus* group, food poisoning, minimum inhibitory concentration, WGS

## Abstract

**Background:** *Bacillus cereus* is a widespread environmental Gram-positive bacterium which is especially common in soil and dust. It produces two types of toxins that cause vomiting and diarrhea. At present, foodborne outbreaks due to *Bacillus cereus* group bacteria (especially *Bacillus cereus* sensu stricto) are rising, representing a serious problem in the agri-food supply chain. **Methods:** In this work, we analyzed 118 strains belonging to the *Bacillus cereus* group, isolated from several food sources, for which *in vitro* and *in silico* antibiotic resistance assessments were performed. **Results:** Many strains showed intermediate susceptibility to clindamycin, erythromycin, and tetracycline, suggesting an evolving acquisition of resistance against these antibiotics. Moreover, one strain showed intermediate resistance to meropenem, an antibiotic currently used to treat infections caused by *Bacillus cereus*. In addition to the phenotypic antimicrobial resistance profile, all strains were screened for the presence/absence of antimicrobial genes via whole-genome sequencing. There was inconsistency between the *in vitro* and *in silico* analyses, such as in the case of vancomycin, for which different isolates harbored resistance genes but, phenotypically, the same strains were sensitive. **Conclusions:** This would suggest that antibiotic resistance is a complex phenomenon due to a variety of genetic, epigenetic, and biochemical mechanisms.

## 1. Introduction

The *Bacillus cereus* group, also named *Bacillus cereus* sensu lato (s.l.), is a heterogeneous group of aerobic or facultative anaerobic bacteria consisting of several phylogenetically correlated species [1,2]. They are Gram-positive, ubiquitous in the environment, and can grow at optimal temperatures ranging from 30 to 40 °C and in a pH range between 5 and 8.8 [3]. Most strains are catalase-positive and mobile [1]. A peculiar characteristic of these microorganisms is their ability to form spores, which are metabolically dormant cell types resistant to extreme conditions including heat, freezing, drying, and radiation (commonly used in the food industry) [4,5].

In the environment, they populate all kinds of soils, waters, sediments, and plants when in their spore form, but they can also be detected in animals [6,7]. Given their pervasiveness in the environment, *Bacillus* spores could contaminate raw food ingredients (vegetables, milk, fruit, spices, and cereals). Thus, a wide variety of processed or ready-to-eat food products might contain these bacteria; additionally, germination and outgrowth during the stockpiling/conservation phase are also possible, causing foodstuff spoilage [8].

The *B. cereus* group consists of several species, and Carroll et al. [9] proposed a new taxonomic nomenclature that included eight genomospecies names: *B. cereus* sensu stricto (*B. cereus* s.s.), *B. mosaicus*, *B. toyonensis*, *B. mycoides*, *B. paramycoides*, *B. pseudomycoides*, *B. luti,* and *B. cytotoxicus*. Among them, *B. cereus* s.s. is the model species of the *B. cereus* group. It can complete a full saprophytic life cycle, but it can also be an opportunistic human pathogen [10] that causes gastrointestinal illness, bacteremia, endocarditis, respiratory and urinary tract infections, endophthalmitis, and meningitis [2,11,12]. Moreover, *B. cereus* s.s. is one of the most common pathogens in food poisoning [13], being responsible for diarrhea and emetic syndrome. This bacterium colonizes raw food but can also affect milk and dairy products due to, for example, pasteurization failure, as well as processed meat as a consequence of nonhygienic slaughtering practices or contaminated food processing environments [14]. *B. cereus* s.s. and related bacteria are considered mainly responsible for foodborne diseases and, in the European Union (EU), *B. cereus* is the fifth most significant etiological agent of foodborne outbreaks [15]. However, *B. cereus* infections are not only reported in the EU. For example, in the United States, about 300 outbreaks occurred between 1998 and 2015 [15], while, in China, it caused 6.8% of outbreaks from 1992 to 2001 [14]. According to the European Food Safety Authority (EFSA), *B. cereus* causes several foodborne illnesses in humans [16]. However, the real number of cases is underestimated; because it is not considered a high-risk microorganism, it is not regularly monitored by competent public health authorities except in response to epidemic outbreaks [15]. Moreover, due to similarity with the clinical symptoms caused by *Staphylococcus aureus* and *Clostridium perfringens* food poisoning [17], *B. cereus* s.s. infection is often misdiagnosed. In particular, *B. cereus* s.s. has been incriminated as a cause of toxin-induced emetic and diarrheagenic syndromes after ingestion [1], which represent the foremost concern for public health services [17], considering the different spectrum of diseases provoked (from gastrointestinal forms that require a short recovery time to more serious systemic diseases like bacteremia and septicemia, which could have a fatal outcome). Generally, *B. cereus*-related infection symptoms start from 0.5 to 16 h after the ingestion of contaminated food and disappear within 24 h without requiring pharmaceutical treatment [18]. Severe *B. cereus* infections are treated with antibiotics, but the excessive and inappropriate use of these molecules could lead to the antibiotic resistance phenomenon [19]. Antibiotic resistance is currently a critical public health issue and global priority, given that infections caused by resistant bacteria are more difficult to treat than those caused by nonresistant ones, resulting in higher medical costs, prolonged hospital stays, and increased mortality [20]. Bacterial resistance to antibiotics can be intrinsic (or innate), i.e., naturally present within the genome of a bacterial species and independent of antibiotic pressure [21], or acquired. The latter situation is due to the acquisition of exogenous genes carried by plasmids, integrons, transposons, and bacteriophages [22]. *B. cereus* strains have generally been found to be resistant to β-lactam antibiotics [23] but are usually sensitive to chloramphenicol, clindamycin, vancomycin, ciprofloxacin, erythromycin, gentamycin, doxycycline, rifampin, linezolid, meropenem, and tetracycline; some strains have even shown intermediate susceptibility [17,24,25,26]. The mechanisms sustaining the antibiotic resistance of *B. cereus* are not completely understood due to the scarce number of studies focusing on it. However, it is known to be a β-lactamase-producing bacterium [12], and Parulekar and Sonawane [27] attempted to study its aminoglycoside resistance by studying the aminoglycoside phosphotransferase of *B. cereus in silico* and *in vitro*.

Therefore, the active surveillance of food contamination by *B. cereus* s.l. and, more specifically, antimicrobial resistance profile detection is necessary to provide new insights into the role of *B. cereus* group strains as reservoirs of antibiotic resistance genes that can be transferred into the food production chain. Thus, the management of infections caused by these bacteria should also include an analysis of the antibiotic susceptibility of the strains and their genetic content.

In this study, we analyzed *B. cereus* group strains isolated from different food sources, aiming to investigate the correlations of their in vitro and in silico antibiotic susceptibility profiles and their phylogenomic relatedness.

## 2. Results

### 2.1. Isolation of B. cereus Group Strains

The isolation of *B. cereus* strains from food samples was assessed via a confirmation of typical color morphology, which is dull gray and opaque with a rough matted surface, irregular perimeters, and zones of hemolysis surrounding the colonies.

### 2.2. MALDI-TOF Mass Spectrometry (MS) Analysis

All analyzed strains were identified as *B. cereus* via MALDI-TOF MS, using the commercial BDAL library (MBT Compass library v 7.0.0.0, Bruker Daltonics GmbH & Co. KG, Bremen, Germany) with a log (score) that generally ranged between 1.7 and 2.0, accompanied by the following comment: “*B. anthracis*, *B. cereus*, *B. mycoides*, *B. pseudomycoides*, *B. thuringiensis* and *B. weihenstephanensis* are closely related and members of the *B. cereus* group. In particular, *B. cereus* spectra are very similar to spectra from *B. anthracis*. *B. anthracis* is not included in the MALDI Biotyper database. For differentiation, an adequate identification method must be selected by an experienced professional. The quality of spectra (score) depends on the degree of sporulation: Use fresh material”.

### 2.3. WGS Analysis

Based on a Btyper ANIblast prediction, the 118 strains were classified as *B. mosaicus* (47 isolates), *B. cereus* s.s. (45 isolates), *B. toyonensis* (7 isolates), *B. cereus* s.s. biovar Thuringiensis (6 isolates), *B. mosaicus* subsp. *cereus* biovar Emeticus (4 isolates), *B. mosaicus* subsp. *cereus* (6 isolates), *B. mosaicus* biovar Thuringiensis (2 isolates), and *B. mycoides* (1 isolate) (Appendix A). The genetic distance among the isolates was calculated using cgMLST profiles. The phylogenetic tree (Figure 1) showed higher genetic diversity among the 118 isolates. The isolates were grouped in three clades (1, 2, and 3) through the *panC*-gene-based group attribution, as proposed by Guinebretière M.-H. et al. [28]: clade 1 (58 isolates), clade 2 (51 isolates), and clade 3 (8 isolates). Clade 3 also included the outgroup *B. citotoxicus*. Specifically, the isolates identified as *B. mosaicus* and *B. mosaicus* subspecies cereus, *B. mosaicus* biovar Thuringiensis, *B. mosaicus* subspecies *cereus* biovar Thuringiensis, and *B. mosaicus* subspecies *cereus* biovar Emeticus were grouped in clade 1; all isolates identified as *B. cereus* s.s. and *B. cereus* s.s. biovar Thuringiensis, *B. mosaicus* subspecies cereus, and *B. mosaicus* subspecies *cereus* biovar Thuringiensis were grouped in clade 2, and the isolates identified as *B. mycoides* and *B*. *toyonensis* were grouped in clade 3. Finally, the cgMLST-based dendrogram showed that there is no association between the identified species and the food source category from which they were isolated.

Antimicrobial resistance (AMR) gene prediction showed that *van*R (98%) was the most abundant gene, followed by *bla*1 and *bla*2 (96%), *van*ZF (88%), and *van*RM (84%); other genes were predicted with an occurrence ≤ 68% (Appendix A). They were grouped into seven categories based on their putative drug class interaction (Figure 2): β-lactamase (*bla*1, *bla*2, *Bc*I, *Bc*II, *bla*P, and *bla*Z_12), glycopeptide (*van*R; *van*RA, *van*RB, *van*RF, *van*RM, *van*R-Pt, *van*S, *van*S-Pt2, *van*YA, *van*YF, *van*Y-Pt, and *van*ZF), macrolide (*mph*A and *mph*B), fluorochinolone (*arl*S), tetracycline (*tet*L), phenicol (*cat*A), and clindamycin (*lsa*B) antibiotics. The presence of at least one gene associated with β-lactamase resistance was predicted in all 118 isolates. No resistance-associated point mutations were detected via an *in silico* AMR protein sequence comparison.

### 2.4. Antibiotic Susceptibility Analysis

After 16 h of incubation at 37 °C, the 96-well plates were read and minimum inhibitory concentration (MIC) values were checked (Table 1 and Appendix A). All 118 strains (100%) exhibited resistance to penicillin G according to the literature [23], while they were sensitive to gentamicin, doxycycline, ciprofloxacin, rifampicin, chloramphenicol, vancomycin, and linezolid. Regarding the meropenem molecule, all strains displayed sensitivity except for one strain (0.9%) that exhibited intermediate susceptibility. Moreover, 3/118 (2.5%) strains displayed intermediate resistance to tetracycline, 8/118 (6.8%) to erythromycin, and 23/118 (19.5%) to clindamycin (Appendix A).

A few strains presented intermediate susceptibility for two antibiotics and were found to be resistant to erythromycin and clindamycin and tetracycline and clindamycin, respectively (Appendix A).

## 3. Discussion

*B. cereus* s.s., as the main member of the *Bacillus cereus* s.l. group, is involved in food contamination. In our study, we described 118 food-derived strains belonging to the *B. cereus* group. Phylogenetic analysis, based on cgMLST, showed a high inter-sample genetic diversity with no relationship to their taxonomical attribution nor source of isolation. Specifically, among the strains, 45 were identified as *B. cereus* s.s.; most of them (28 isolates) were isolated from milk and dairy products (e.g., “mozzarella”, “scamorza”, and “ricotta”), followed by vegetables and pastry/bakery products (e.g., flour, biscuits, and sweets), meat, gastronomic preparations, fish, and pasta. Meat and fish were also contaminated with *B. cereus* s.s., although to a lesser extent.

*B. cereus* s.s. was also found in mixed food, e.g., a meat sandwich, Russian salad, or focaccia with scamorza cheese, making the exact determination of the contaminated ingredient and the supply chain step at which contamination may have occurred quite difficult. Other members of the *B. cereus* group identified in food matrices were *B. mosaicus*, *B. mosaicus* biovar Thuringiensis, *B. cereus* biovar Thuringiensis, *B. mosaicus* biovar Emeticus, *B. mycoides,* and *B. toyonensis*. The abovementioned *Bacillus* species are not common human pathogens; instead, they are useful in the agricultural field. For example, *B. mosaicus* biovar Thuringiensis and *B. toyonensis* are used as biocontrol agents, while *B. mosaicus* and *B. mycoides* seem to promote plant growth [11,31,32,33,34]. Even if the risk these bacteria pose to human health is underestimated, they can represent potential food hazards as they may cause food poisoning because some strains, e.g., *B. cereus* s.s., are able to produce toxins and virulence factors [35,36,37]. Therefore, the investigation of strain antimicrobial sensitivity or resistance would allow for a better definition of specific therapeutics. It is known that bacteria can acquire antibiotic resistance genes (ARGs) through horizontal gene transfer (HGT) by means of different types of genetic elements (plasmids, integrons, and transposons) and different modalities of transfer (transformation, conjugation, and transduction) [38]. Transfer can occur between the same species or between different species, and it is also possible that residual DNA carrying ARGs may persist in the environment for a long time even after the death of resistant strains, with the possibility of transferring them to other strains [39]. This is why the investigation of antimicrobial susceptibility is also of interest for bacteria that are not highly pathogenic or are nonpathogenic, such as members of the *B. cereus* group. All 118 strains exhibited phenotypic sensitivity to gentamicin, doxycycline, ciprofloxacin, rifampicin, chloramphenicol, vancomycin, and linezolid, as supported by previous studies [17,24,25,40]. Gentamicin, vancomycin, and linezolid are commonly used as therapeutic treatments in severe *B. cereus* infections [41], and our results confirmed the efficacy of these antibiotics. Treatment with clindamycin is also frequent in case of *B. cereus* infection [41]. In our study, 23 strains (18 *B. cereus* s.s., 3 *B. cereus* biovar Thuringiensis, and 2 *B. mosaicus*) (20,3% of total strains) showed phenotypic intermediate susceptibility to clindamycin, which is in agreement with other studies [24,25,41]. The increasing number of strains with intermediate susceptibility could be explained as a general progressive acquisition of resistance by members of the *B. cereus* group against clindamycin [40]. Furthermore, eight strains (four *B. mosaicus*, one *B. mosaicus* subsp. *cereus*, one *B. cereus* s.s., one *B. mosaicus* biovar Thuringensis, and one *B. toyonensis*), corresponding to 6,8% of strains, exhibited phenotypic intermediate susceptibility to erythromycin, according to the current literature [25,42,43], while a smaller number of strains (two *B. mosaicus* and one *B. cereus* s.s, corresponding to 2,5% of strains) expressed intermediate susceptibility to tetracycline, as previously reported [24,44,45]. We also observed an interesting intermediate behavior of a single strain of *B. mosaicus* against meropenem, an antibiotic often used in clinical practice. Meropenem belongs to the carbapenem drug class, β-lactam antibiotics, which are active against many aerobic and anaerobic Gram-positive and Gram-negative bacteria. It is active against extended-spectrum β-lactmases [46] but may be more susceptible to metallo-β-lactamases. Its intermediate susceptibility is probably due to the presence of the *bla* genes, which have penicillin-, cephalosporin-, and carbapenem-hydrolyzing activities [47]. No association between the in silico predicted ARGs (Appendix A) and phenotypic characteristics (Appendix A) was found. For example, most of the *B. cereus* group strains carried vancomycin resistance genes, but all of them were found to be susceptible to this antibiotic.

All eight strains with intermediate resistance to erythromycin carried genes responsible for resistance to this molecule, but other strains did not express an intermediate or resistant phenotype despite possessing some resistance genes against the macrolide class.

Analogously, only three strains harbored the *tetL* gene, which is responsible for tetracycline resistance, and the *in vitro* analysis showed intermediate susceptibility to this drug. Inconsistency between phenotypic and genetic data is not surprising considering the so-called “silent resistome” [48,49,50]. The term “resistome” denotes all genes capable of conferring single or multiple antimicrobial resistance [51] and includes constitutively expressed genes, precursors, and acquired AMR genes [50]. The concept of the “silent (also defined as “cryptic”) antimicrobial resistance gene” is also intriguing. These genes may consist of plasmid or chromosomal CDSs that are not canonically expressed or have a very low expression level, causing bacterial sensitivity to specific drugs [49,50]. Usually, transcription/translation for multiple AMR genes is not constitutively active due to the metabolic cost of these pathways. Basically, the down-regulation or complete silencing of AMR genes would contribute to preserving microbial fitness. Resistance gene silencing is a phenomenon that has been recorded in the literature and evaluated in certain species such as *E. coli, Salmonella* and *Staphylococcus aureus* [52,53,54]. Gene silencing could occur in several ways; it could be linked to a specific expression regulation signal [52], to the insertion of integrons into the operon [53], or even to mutations in a gene sequence [54]. Various examples and theoretical hypotheses of a silent resistome are described in the literature [48]. Other explanations are also plausible, such as horizontal gene transfer, regulatory mechanisms, and environmental influences [51]. At present, it is unclear which molecular mechanisms play a fundamental role in silencing the predicted AMR genes. These aspects have not been addressed here, and further studies could help us understand this phenomenon. In addition, we observed that a very low percentage (1.7%) of strains (2/118) exhibited a simultaneous intermediate susceptibility against two different antibiotics, suggesting that, in members of the *B. cereus* group, a progressive acquisition of multidrug resistance could also occur, leading to antibiotic treatment failure.

## 4. Materials and Methods

### 4.1. B. cereus Group Isolated Strains

A total of 118 *B. cereus* group strains analyzed in this study were isolated from different types of food (dairy products, vegetables and fruit, fish and meat, and bakery products) collected during routinary food surveillance among different production and distribution sites in the Apulia and Basilicata regions.

Isolation was performed as previously described [55]. Briefly, 5 g of food matrix was added to 45 mL of Buffered Peptone Water (BPW) (Biolife Italiana, Milan, Italy) and homogenized using a stomacher (PBI International, Milan, Italy) at 230 rpm for 30 s. Then, 45 mL of double-strength Tryptone Soy Polymyxin Broth (TSPB) (Biolife Italiana, Milan, Italy) was added, and the samples were incubated at 30 °C for 48 h. After incubation, 10 mL of enrichment broth was streaked onto the surface of solid selective medium Mannitol Egg Yolk Polymyxin Agar (MYP) (Biolife Italiana, Milan, Italy), and the plates were incubated at 30 °C for 24–48 h under aerobic conditions. Then, typical presumptive *Bacillus cereus* group colonies for each sample were selected and subcultured on Columbia Agar with 5% sheep blood.

### 4.2. MALDI-TOF Mass Spectrometry Analysis

As the first screening, Matrix-Assisted Laser Desorption/Ionization Time-Of-Flight Mass Spectrometry (MALDI-TOF MS) analysis was performed to quickly identify the *Bacillus cereus* group bacteria [55]. After incubation at 37 °C for 18–24 h, the bacterial isolates were subjected to MALDI-TOF MS analysis. The bacterial colony was directly transferred using a toothpick on the 96-well target plate (Bruker Daltonics GmbH & Co. KG, Bremen, Germany), covered with 1 μL of 70% formic acid, dried at room temperature, and then overlaid with 1 μL of an α-cyano-4-hydroxycinnamic acid (CHCA) matrix solution (a saturated solution of α-cyano-4-hydroxycinnamic acid in 50/50 (*v*/*v*) acetonitrile/H_2_O containing 2.5% trifluoroacetic acid) and allowed to dry. Mass spectra were acquired using a Microflex LT/SH™ mass spectrometer (Bruker Daltonics GmbH & Co. KG, Bremen, Germany) operating in linear positive mode and covering a mass range between 2 and 20 kDa. For the validation of runs, the instrument was calibrated using Bruker Bacterial Test Standard (BTS, Bruker Daltonics GmbH & Co. KG, Bremen, Germany) and an extract of the *Escherichia coli* DH5α strain spiked with two additional pure proteins (RNase A of 13,683.2 Da and myoglobin of 16,952.3 Da).

### 4.3. WGS Sequencing and Bioinformatic Analyses

Typing analysis of the 118 strains was performed via WGS since MALDI-TOF MS is not definitively exhaustive for the identification of the bacteria belonging to the *Bacillus cereus* group. DNA extraction was performed using a DNAeasy Blood and Tissue kit (Qiagen, Hilden, Germany), following the manufacturer’s protocol for Gram-positive bacteria. Extracted DNA quality and concentrations were estimated by Qubit 3.0 using Qubit dsDNA HS Assay (Thermo Fisher Scientific, Waltham, MA, USA). Libraries were prepared using the DNA Prep Illumina (Illumina, San Diego, CA, USA) and sequencing was performed. For each isolate, paired-end genomic libraries were prepared using DNA Prep Illumina. Sequencing was performed on the Illumina MiSeq platform using 500 cycle chemistry (2 × 250 paired-end reads). The raw data were trimmed using Trimmomatic (Galaxy Version 0.36.6) [56] and the draft genomes were then generated using the Unicycler v.0.5.0 [57] de novo assembler with default parameters. The draft genomes were submitted to BTyper [58], as previously described by Bianco and colleagues [24]. Several versions of BTyper were used: BTyper v1.0 and v2.3.2 were used for AMR gene prediction and BTyper3 [59] v3.2.0 was used for the taxonomic classification of the *B. cereus* group based on pairwise genomic similarity, as calculated via the integrated method Average Nucleotide Identity BLAST (ANIBlast). Phylogenetic group prediction based on the *panC* gene was the same regardless of the version used. The AMR genes were predicted using the software ABRicate v0.8, which includes different predownloaded databases: ARG-ANNOT [60], NCBI AMRFinderPlus [61], CARD [62], ResFinder [63], and PlasmidFinder [64]. The known antimicrobial resistance point mutations along the detected AMR genes were searched by means of the Resistance Gene Identifier (RGI) tool, which is available on the CARD database (https://card.mcmaster.ca/analyze/rgi, accessed on 16 July 2024), maintaining default parameters. Core genome Multi-Locus Sequence Typing “cgMLST” for the draft genomes was performed using the scheme (1568 loci) developed by Tourasse N. et al. [65], which is available on PubMLST (https://pubmlst.org/, accessed on 10 July 2024). Subsequently, sample-specific cgMLST allelic profiles were pairwise-compared by means of the ChewTree tool (https://aries.iss.it/, accessed on 11 July 2024), while the resulting Newick-formatted phylogram was used to generate a neighbor-joining tree through Microreact (https://microreact.org/, accessed on 11 July 2024). The identified draft genome assemblies of the *B. cereus* group were deposited in GenBank under BioProject PRJNA826696, while the genome sequences and metadata were deposited in PubMLST (“*Bacillus cereus* isolates database”). PubMLST strain identifiers are reported in Appendix A.

### 4.4. Antimicrobial Susceptibility Tests

The microdilution method was used to verify the susceptibility of each isolate to the main antibiotic classes (β-lactam, aminoglycosides, tetracycline, fluoroquinolones, rifamycin, chloramphenicol, macrolides, lincosamides, glycopeptides, carbapenem, and oxazolidinones), according to Clinical and Laboratory Standard Institute guidelines [29,30]. The antibiotics and related concentrations used for the microdilution experiments were as follows: penicillin G (0.031–4 µg/mL), gentamicin (0.125–16 µg/mL), doxycycline (0.25–32 µg/mL), ciprofloxacin (0.25–32 µg/mL), rifampin (0.25–32 µg/mL), chloramphenicol (1–128 µg/mL), erythromycin (0.125–16 µg/mL), tetracycline (0.125–16 µg/mL), clindamycin (0.125–16 µg/mL), vancomycin (0.125–16 µg/mL), meropenem (0.125–16 µg/mL), and linezolid (0.125–16 µg/mL). Antibiotic powders were resuspended in an appropriate solvent and 100 µL of the solution was then pipetted into 96-well plates. Scalar dilutions of each antibiotic were established in a range of values included in the reference breakpoints.

After overnight incubation on Columbia Agar with 5% sheep blood, 1–2 bacterial colonies were suspended in a 0.5 McFarland standard sterile saline solution. The bacterial suspensions were further diluted 1:100 in cationic-adjusted Muller–Hinton broth (CAMHB) and then inoculated into 96-well plates containing specific antibiotics. Bacterial growth was detected after 16 h of incubation at 37 °C. The MIC, which corresponds to the lowest antibiotic concentration able to inhibit bacterial growth, was detected for each antibiotic. The obtained MIC values were interpreted using CLSI breakpoints [29,30].

The CLSI breakpoints (mg/mL) for penicillin G, meropenem, vancomycin, gentamicin, erythromycin, tetracycline, ciprofloxacin, clindamycin, chloramphenicol, and rifampin were those suggested for *Bacillus* spp. (not *Bacillus anthracis*), according to CLSI document M45 [30], whereas for linezolid and doxycycline, interpretative criteria for *Staphylococcus* spp. were used according to CLSI M100 [29]. *Staphylococcus aureus* ATCC 29213 and *E. coli* ATCC 25922 were used as control strains to check experimental validity.

## 5. Conclusions

*B. cereus* infection is far more frequent than expected, and new data suggest its involvement in food contamination; however, it is plausible that the incidence of related foodborne illnesses is likely underestimated because the symptoms are usually weak and self-limiting [26,66]. In this study, 118 isolates of the *B. cereus* group were detected in several food typologies. We tested them for their susceptibility to the most common antibiotics used in clinical practice. *In vitro* analysis showed widespread susceptibility to most of the antibiotics tested, but intermediate resistance was detected in some cases. Moreover, WGS analysis should be performed simultaneously to detect the possible presence of strains carrying AMR genes. We found that the presence of these genes in the genome of the strains analyzed is not always correlated to a phenotypic antimicrobial resistance, as in the case of vancomycin. These strains can pose a major public health problem because they constitute a reservoir of ARGs that could be transmitted via HGT between different species and genera. They can also become active in cell hosts, resulting in a new resistance phenotype [48,49,50]. In conclusion, we suggest that both phenotypical and genetic analyses are important and complementary to correctly interpret the antibiotic resistance of bacteria. Their combined use could lead to an improvement in medical treatment strategies and in the selection of the most appropriate antibiotics in order to avoid failure of antibiotic therapy and the possible development of the antimicrobial resistance phenomenon.

## Figures and Tables

**Figure 1 antibiotics-13-00898-f001:**
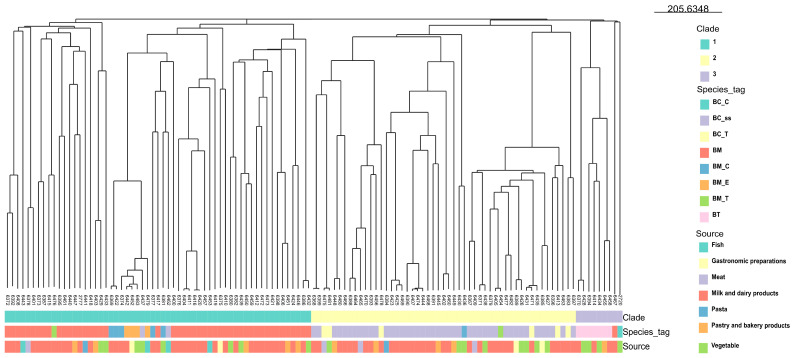
Phylogenetic tree based on cgMLST allele profiles from 118 *B. cereus* group isolates. The phylogenetic tree was constructed using the neighbor-joining (NJ) method. The isolates were indicated with pubMLST ID (Appendix A). The clade, species, and source were added and color-coded, as reported in the legend. *B. cytotoxicus* NVH 391-98 (pubMLST ID: 2720) was used as an outgroup. BC_ss: *B. cereus* s.s.; BM: *B. mosaicus*; BC_T: *B. cereus* s.s. biovar Thuringiensis; BM_C: *B. mosaicus* subsp. *cereus*; BM_T: *B. mosaicus* subsp. *cereus* biovar Thuringiensis; BM_E: *B. mosaicus* subsp. *cereus* biovar Emeticus; BT: *B. thuringiensis*; BC_C: *B. cytotoxicus*.

**Figure 2 antibiotics-13-00898-f002:**
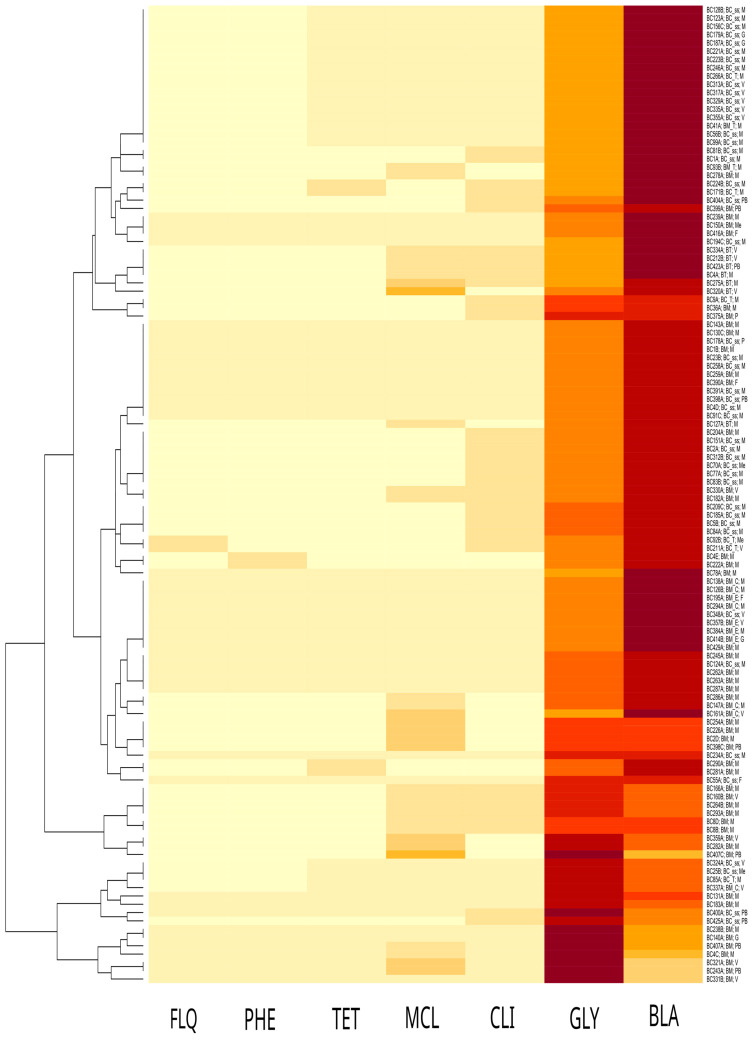
Heatmap of antibiotic resistance gene occurrence (clustered by antibiotic molecule class) among strains. Colors range from light yellow (n. of predicted genes for a drug class = 0) to dark red (n = 10); *x*-axis: antibiotic molecule class; *y*-axis (left side): strains hierarchical clustering by gene occurrence; *y*-axis (right side): strain ID; species nomenclature (“BC_ss”: *B. cereus* s.s.; “BM”: *B. mosaicus*; “BC_T”: *B. cereus* s.s. biovar Thuringiensis; “BM_C”: *B. mosaicus* subsp. *cereus*; “BM_T”: *B. mosaicus* subsp. *cereus* biovar Thuringiensis; “BM_E”: *B. mosaicus* subsp. *cereus* biovar Emeticus); source of isolation (“M”: milk and dairy products; “G”: gastronomic preparations; “Me”: meat; “V”: vegetable; “P”: pasta; “F”: fish; “PB”: pastry and bakery products); antibiotic drugs (“FLQ”: fluoroquinolone; “PHE”: phenicol; “TET”: tetracycline; “MCL”: macrolide; “CLI”: clindamycin; “GLY”: glycopeptide; “BLA”: β-lactam).

**Table 1 antibiotics-13-00898-t001:** MIC value results of antibiotic susceptibility tests.

Drug Class	Antibiotic	MIC Breakpoints (µg/mL) **			MIC Value Results (%)		
		S * (≤)	I *	R * (≥)	S *	I *	R *
Beta-lactam	Penicillin G	0.12	*na ****	0.25	*na*	*na*	100
Beta-lactam	Meropenem	4	8	16	99.1	0.9	*na*
Aminoglycoside	Gentamicin	4	8	16	100	*na*	*na*
Tetracycline	Doxycycline	4	8	16	100	*na*	*na*
Tetracycline	Tetracycline	4	8	16	97.5	2.5	*na*
Fluoroquinolone	Ciprofloxacin	1	2	4	100	*na*	*na*
Rifamycin	Rifampicin	1	2	4	100	*na*	*na*
Amphenicol	Chloramphenicol	8	16	32	100	*na*	*na*
Macrolide	Erythromycin	0.5	1–4	8	93.2	6.8	*na*
Lincosamide	Clindamycin	0.5	1–2	4	80.5	19.5	*na*
Glycopeptide	Vancomycin	4	*na*	*na*	100	*na*	*na*
Oxazolidinone	Linezolid	4	*na*	8	100	*na*	*na*

* Sensitivity (S), resistance (R), and intermediate (I) susceptibility. ** Interpretation criteria for MIC according to Clinical and Laboratory Standard Institute guidelines [29,30]. *** no breakpoint available.

## Data Availability

The genomic data presented in this study are openly available in GenBank (BioProject PRJNA826696). GenBank (https://www.ncbi.nlm.nih.gov/bioproject/?term=PRJNA826696) (accessed on 2 October 2022). BioProject PRJNA826696. Additionally, the data are deposited in the PubMLST resource (“*Bacillus cereus* isolates database”).

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
