# Peer review of "Antimicrobial and Phylogenomic Characterization of Bacillus cereus Group Strains Isolated from Different Food Sources in Italy"

_antibiotics, 2024, doi:10.3390/antibiotics13090898_

Round 1

Reviewer 1 Report

Comments and Suggestions for Authors

The manuscript on the antimicrobial characterization of Bacillus cereus group strains from the Apulia and Basilicata regions is important for advancing scientific knowledge, improving public health and food safety, guiding agricultural practices, and fostering further research and collaboration. Such work has significant implications for both local and broader contexts, contributing to safer food systems and enhanced understanding of microbial behavior in specific environments. Antibiotic resistance in Bacillus cereus exemplifies the silent pandemic of antimicrobial resistance, posing significant threats to public health, food safety, and global economies. Addressing this issue requires comprehensive strategies that include surveillance, education, prudent antibiotic use, and innovative research. By understanding and tackling the silent pandemic, we can mitigate its impact and safeguard the effectiveness of antibiotics for future generations. Thank you for the contribution. 

Comments on the Quality of English Language

Author Response

Thank you for reviewing our paper and for your valuable opinion and comments. However, the manuscript has now been revised following the other reviewers’ comments.

Reviewer 2 Report

Comments and Suggestions for Authors

Abstract

The conclusion in the abstract needs some adjustments for clarity and accuracy. While the analysis showing an incomplete correlation between antimicrobial genes and phenotypic resistance is intriguing, the statement about the 'inappropriate use of antibiotics' needs more specific backing. I suggest clarifying what other factors might be contributing to resistance in these strains. Additionally, ensure that the conclusion directly reflects the results of your study without overgeneralizing.

Introduction

1. Clearly state what is not known or what is insufficiently studied about Bacillus cereus group strains, particularly in the context of antimicrobial resistance. Emphasize the lack of comprehensive data on antibiotic resistance mechanisms among these species.

2. Author did not include epidemiological data on the prevalence of infections caused by Bacillus cereus group species.

3. Author did not present data on the trends in antibiotic resistance among these species.

Method

1. The WGS data appears to be underutilized in the current study. Phylogenetic tree can be constructed. ANI analysis should performed

Result

1. Fig 1 is poorly presented. Author can present only the presence of genes. Heat map of virulence and antibiotic resistance genes should performed (not in the supplementary)

2. Fig 2 should be changed to table.

Discussion

1. If you choose to discuss the silent resistome hypothesis, please provide more context and evidence for this phenomenon. Cite relevant studies that support this hypothesis and explain how it applies to your findings in the Bacillus cereus group strains.

2. There are other explanations as well such as horizontal gene transfer, regulatory mechanisms and environmental influences,

Reviewer 3 Report

Comments and Suggestions for Authors

In the study entitled ''Antimicrobial characterization of Bacillus cereus group strains  isolated from different food sources in Apulia and Basilicata  regions'', the authors analyzed different food sources for the presence of bacteria  belonging to Bacillus cereus group to investigate the presence of these in different food  sources. They have performed WGS analysis of 118 strains and in silico prediction of antimicrobial resistance genes, and finally examined the correlation of the presence of antimicrobial resistance genes and their in vitro sensitivity to antimicrobials. This is an interesting survey showing that molecular  methods like WGS analysis should be simultaneously performed to detect strains carrying “silent antimicrobial resistance genes”, which represent a reservoir of ARGs. The methodology and the presentation of the results need major revisions,  and the manuscript needs extensive editing in English  bebore it is accepted for publication. In particular,

Major comments:

- Title: '' in Apulia and Basilicata  regions '' is not necessary information for the title and may be ommited. Please replace with ''in Italy''.

- A phylogenomic analysis of the 118 is missing. A SNP core genome analysis or cgMLST analysis (please see Nicolas J. Tourasse, Keith A. Jolley, Anne-Brit Kolstø, Ole Andreas Økstad,Core genome multilocus sequence typing scheme for Bacillus cereus group bacteria,Research in Microbiology,Volume 174, Issue 6,2023,104050,ISSN 0923 2508,https://doi.org/10.1016/j.resmic.2023.104050. https://www.sciencedirect.com/science/article/pii/S0923250823000256)  would provide further information about the genomic relatedness of the isoltes recovered from different samples.

- Besides the presence/absence of AMR genes, did the authors examine the presence of mutations in the AMR genes that could lead in loss of function of the respective gene products and thus sensitivity to antimicrobial agents?

- The conclusions should be descriptive of the major  findings of this study and not general information and references of previous studies. Thus, lines 328 -342 may be transferred in the discussion section. Please be more specific on the conclusions from the results of this study.

Minor comments:

- lines 23-28: please rephrase these sentences for clarity

- lines 50-71: may be presented in one paragraph. 

- line 58: please use plural ''insects''

- line 59: please replace ''was found involved'' with ''was the causative agent''

- line 66: please replace ''Concerning B. wiedmanni, it is'' with ''B. wiedmanni is''

- line 73: please replace ''originates an important number of'' with ''causes several''

- line 74: please delete ''at the date''

- line 76: please replace the ''foremost worries'' with ''foremost concern''

- lines 93-94:  please replace ''and we evaluated in vitro their sensitivity to antimicrobials also correlating these results with genetic analyses'' with ''and we evaluated the correlations of the in vitro sensitivity to antimicrobials with the presence/absence of AMR genes. 

- lines 103-104: Please add B. in italics before each species, such as B. cereus, B. mycoides, B. pseudomycoides, B. thuringiensis and B. weihenstephanensis.

- line 111: ''7 B. toyonensis'' please correct ''B.'' in italics. 

- line 117. please s.l.: define in the first appearance  

- lines 127,198: please replace ''antimicrobic '' with antimicrobial''

- line 129: please replace ''β-lactams'' with ''β-lactam''

- lines 132-133: please rephrase as '' Of 118 isolates, 2 (2%) carried fluoroquinolone resistance  (arlS) or chloramphenicol resistance (catA) genes.

- lines 125, 170, 219, 301,  please replace ''behavior'' with ''susceptibility''

- line 162: What do the authors mean with the word ''contemporary''- is there such a term for antimicrobial resistance?

- lines 187-188: Please correct the grammar of this sentence. Do the authors mean ''Bacillus species mentioned here in are not common human pathogens; instead, they are useful in the agricultural field (please give some examples).

- lines 207-208: please rephrase this sentence for clarity and explain better this statement based on the cited references\

- line 217: please ''works'' with ''is active''

- line 220: please replace ''abilities'' with ''activities''

- line 224: please replace ''resulted'' with ''were''

- line 227: please use olural ''genes''

- line 271: please correct as ''transferred''

-

Comments on the Quality of English Language

The manuscript needs extensive editing in English 

Round 2

Reviewer 3 Report

Comments and Suggestions for Authors

The authors have successfully revised the manuscript according to the recommendations. Minor changes are needed before publication.

- Title: please add the phylogenomic analysis, which was performed in the revised manuscript, such as: ''Antimicrobial and phylogenomic characterization of Bacillus cereus group strains isolated from different food sources in Italy''

-  Please reform in a single paragraph lines: 48-69,  70-82, 83-101, 171-182, 220-252, 253-276, 279-293.

- Please write the genus Bacillus in full on first appearance, and then replace with B. in italics throughout the manuscript, eg. correct as: B. anthracis, B. cereus, B. mycoides, B. pseudomycoides, B. thuringiensis and B. weihenstephanensis (lines 113- 114), B. cereus (lines 115-116), B. anthracis (line 116). 

- line 25: please replace ''A not complete'' with  ''A non-complete''

- line 48: please replace Carroll et collaborators with Carroll et al.

- line 50: B. paramycoides; please correct B. in italics. 

- lines 102-104: please correct and add the phylogenomic analysis, such as: ''In this study, we analyzed B. cereus group strains isolated from different food sources with the aim to investigate the correlations of the in vitro and in silico antibiotic susceptibility profiles, and their phylogenomic relatedness.

- line 128: please add the authors for reference [28], such as: ''Guinebretière M.-H. et al. [28]''

- line 374: please add a coma and correct ''In this study...'' as ''In this study,...''.

- line 377: please delete ''also', which' is a redundancy in this sentence.

Comments on the Quality of English Language

 Minor editing of English language required
